# The Role of Oxidative Stress in Vitiligo: An Update on Its Pathogenesis and Therapeutic Implications

**DOI:** 10.3390/cells12060936

**Published:** 2023-03-19

**Authors:** Wei-Ling Chang, Chi-Hsiang Ko

**Affiliations:** 1International Ph.D. Program for Cell Therapy and Regeneration Medicine, College of Medicine, Taipei Medical University, Taipei 110, Taiwan; 2School of Pharmacy, College of Pharmacy, Taipei Medical University, Taipei 110, Taiwan

**Keywords:** autoimmune skin disorder, vitiligo, reactive oxygen species (ROS), melanocyte, autoantigen

## Abstract

Vitiligo is an autoimmune skin disorder caused by dysfunctional pigment-producing melanocytes which are attacked by immune cells. Oxidative stress is considered to play a crucial role in activating consequent autoimmune responses related to vitiligo. Melanin synthesis by melanocytes is the main intracellular stressor, producing reactive oxygen species (ROS). Under normal physiological conditions, the antioxidative nuclear factor erythroid 2-related factor 2 (Nrf2) pathway functions as a crucial mediator for cells to resist oxidative stress. In pathological situations, such as with antioxidant defects or under inflammation, ROS accumulate and cause cell damage. Herein, we summarize events at the cellular level under excessive ROS in vitiligo and highlight exposure to melanocyte-specific antigens that trigger immune responses. Such responses lead to functional impairment and the death of melanocytes, which sequentially increase melanocyte cytotoxicity through both innate and adaptive immunity. This report provides new perspectives and advances our understanding of interrelationships between oxidative stress and autoimmunity in the pathogenesis of vitiligo. We describe progress with targeted antioxidant therapy, with the aim of providing potential therapeutic approaches.

## 1. Introduction

Vitiligo is an autoimmune skin disorder characterized by patches of skin that lose pigments because of dysfunctional pigment-producing melanocytes being attacked by immune cells (Figure 1A). The global prevalence of vitiligo is between 0.5% and 2% worldwide, without six biases [1] (0.76% (1.9 million cases in 2020) and 1.11% (2.8 million cases in 2020) in the US [2]). The vitiligo phenotype can manifest as different types, medically referred to as segmental vitiligo (SV) and non-segmental vitiligo (NSV) [1]. Individuals with segmental vitiligo exhibit depigmented patches in only one segment of the body such as the arm, leg, neck, etc. On the other hand, non-segmental vitiligo results in non-symmetrical depigmented patches distributed throughout the individual’s body. Vitiligo greatly affects patients’ quality of life and self-esteem. It also predisposes them to a higher risk of sunburn and skin cancer. The etiology of vitiligo is considered a multifactorial disorder that has been implicated in melanocyte dysfunction due to a combination of genetic susceptibility, generation of inflammation, and autoimmune responses [3]. Among these, oxidative stress is considered to play a crucial role in activating consequent autoimmune responses related to vitiligo.

Reactive oxygen species (ROS) include O_2_^−^, -OH, and H_2_O_2_, which are the main species for evaluating oxidative stress levels. ROS can be produced by cellular metabolic processes or by exogenous exposure. In addition to mitochondrial metabolism, melanin synthesis is one of the major intracellular stressors that produce ROS. Among cellular processes, melanin synthesis behaves as a double-edged sword; it acts as a photoprotector that prevents ultraviolet (UV)-induced DNA damage, but on the other hand, it functions as a photosensitizer that generates high levels of intracellular ROS. In normal physiological conditions, small amounts of ROS are converted into nontoxic substances by the antioxidant system. Overproduction of ROS can occur in pathological situations such as inflammation, or in subjects with a genetic predisposition, which can cause cell and tissue damage [4,5,6]. Antioxidant defenses can be overwhelmed during sustained inflammation such as in chronic disease, neurodegenerative disorders, and aging [5]. Various inflammation-inducing stimuli including TNF-α (tumor necrosis factor-α), lipopolysaccharide (LPS), and thrombin, influence ROS production through nicotinamide adenine dinucleotide phosphate (NADPH) oxidase (NOX) and the mitochondria [7]. Higher levels of intracellular ROS sequentially induce susceptibility to skin disorders [8]. The detection of oxidative stress in patients can provide valuable insights into their disease status and help guide patient management strategies. Oxidative stress can be measured indirectly by measuring the levels of DNA/RNA damage, lipid peroxidation, and protein oxidation from patients’ plasma or serum through fluorescent probes and chemiluminescence assays [9]. Oxidative stress can also be measured by enzymatic antioxidant activities such as superoxide dismutase (SOD), catalase (CAT), gluetathione S-transferases, and glutathione peroxidase [9]. Current studies have shown high levels of antioxidant activities in active vitiligo cases as compared to stable vitiligo and healthy controls by measuring the levels of the antioxidant enzymes SOD and CAT [9,10]. One study provided evidence that the serum oxidative stress indicator, total antioxidant capacity (TAC), malondialdehyde (MDA), and 8-hydroxy-2′-deoxyguanosine (8-OHdG), can indicate the activity and severity in patients with non-segmental vitiligo [11]. Such measurements of oxidative stress in patients’ skin patch areas and blood provide a precise diagnosis and further treatment for patient management. This report summarizes events at the cellular level under excessive ROS in skin from vitiligo.

Oxidative stress is believed to be one of the most crucial initiators of vitiligo. Although the key factors of ROS generation in vitiligo are still not completely understood, several candidates may be considered. An accumulation of hydrogen peroxide causes changes in the antioxidant machinery such as a decrease in catalase levels and increases in superoxide dismutase and other antioxidant enzyme levels [12]. Glutathione (GSH) acts as the first line of defense to eliminate ROS. When GSH is depleted, glutathione reductase 4 (GPX4) is inactivated, resulting in an oxidation-antioxidant imbalance [13]. In the skin lesions of vitiligo, the expression of GPX, catalase (CAT), and methionine sulfoxide reductase (MSR) A and B, as well as other antioxidant enzymes, is downregulated [14]. Overall, these imbalances in the antioxidant system have been demonstrated in vitiligo.

At the cellular level, defective mitochondrial structure and function, such as loss of transmembrane potential, deregulation of cardiolipin and electron transport chain proteins, increased cholesterol in the inner membrane of mitochondria, and increased mass, have been observed in vitiligo [15,16]. Superoxide dismutase 2 (SOD2), a mitochondrial form of SODs that converts O_2_^−^ radicals into H_2_O_2_, has shown increasing activity in vitiligo patients, indicating its role as a genetic risk factor for susceptibility and progression of vitiligo [17]. It is also associated with an unfolded protein response (UPR) in the endoplasmic reticulum (ER) and adhesion defects in the epidermis. Such responses lead to functional impairment and the death of melanocytes, which sequentially initiates an autoimmune response and increases melanocyte cytotoxicity [13].

It is accepted that accumulated ROS in melanocytes consequently provide autoantigens and activate innate and adaptive immune responses in vitiligo [18,19]. The mechanistic relationship between oxidative stress and adaptive immunity was unrevealed until the concept that innate immunity mediating the presence of autoantigens to trigger autoimmunity was raised [3,20]. The presence of autoantigens from stressed melanocytes activates dendritic cells (DCs) and macrophages, which is mediated through the activation of cytotoxic T lymphocytes (CTLs). Herein, we provide new perspectives on advances in the understanding of the pathogenesis of vitiligo, and we attempt to find more interrelationships between oxidative stress and autoimmunity.

## 2. ROS and Melanogenesis

The sources of ROS overproduction can come from both extrinsic and intrinsic factors. From an external perspective, oxidative stress can be stimulated by exposure to the environment such as UVA and UVB, cytotoxic chemicals (e.g., monobenzone and phenols), and medical applications (e.g., certain drugs and hormones). Oxidative stress can also be attributed to internal stressors. A series of cellular metabolic processes, such as mitochondrial metabolism, cellular proliferation, cellular apoptosis, cellular differentiation, and immune reactions can create oxidative byproducts [21,22].

Melanosomes are organelles within melanocytes that synthesize and store melanin, a pigment that protects the skin from UV radiation. Melanin molecules absorb UV radiation and prevent damage to the skin. However, high levels of intracellular ROS are produced as a side product during the process of melanogenesis. This makes melanocytes relatively vulnerable to oxidative stress [23].

In mammals, the melanogenesis pathway is divided into two processes with different catalyzed enzymes for the final types of melanin: eumelanin or pheomelanin (Figure 1B, right panel). Both are initiated by hydroxylation of phenylalanine into L-tyrosine. L-Tyrosine is then hydroxylated and further oxidized to dihydroxyphenylalanine DOPA quinone (DQ). Hydroxylation of L-tyrosine is the rate-limiting step in melanin synthesis and is catalyzed by tyrosinase. After the formation of DQ, the pathway diverges into two pathways leading to the synthesis of reddish-yellow pheomelanin that requires the presence of cysteine, or brownish-black eumelanin due to adding an amino group to DQ with tyrosinase-related proteins 1 (TYRP1) and 2 (TYRP2) or dopachrome tautomerase (DCT). In the melanogenesis process, the activation of redox exchange is required, and oxidative byproducts are produced. This cause melanocytes to be victims of oxidative stress because of the pro-oxidant state generated during melanogenesis. A schematic representation of the melanin synthesis pathway is presented in Figure 1B.

There are several factors that regulate melanogenesis including signaling pathways, micro (mi)RNAs [24], and inflammatory factors [25]. As a master regulator of melanogenesis, the microphthalmia-associated transcription factor (MITF) plays a critical role in the translation of several pigment-specific enzyme genes including tyrosinase, *TYRP1*, and *DCT* (Figure 1B). It was reported that the activated form of phosphorylated MITF is regulated by several kinases such as mitogen-activated protein kinase (MAPK), cAMP-dependent protein kinase (PKA), and glycogen synthase kinase (GSK)-3β, in response to specific environmental stimulants [26]. Understanding the regulatory mechanisms during melanogenesis would improve our ability to treat melanocyte death in vitiligo caused by oxidative stress.

## 3. Antioxidative Defense and Cellular Responses to ROS Accumulation

### 3.1. The Antioxidant, Nuclear Erythroid 2-Related Factor 2 (Nrf2), Pathway

Under oxidative stress, the Nrf2 signaling pathway reacts first to function as an antioxidant. Nrf2 is a transcription factor that can be activated by increased ROS activation, and it translocates to nuclei to regulate the expression of groups of detoxifying and antioxidant defense genes [27]. The activity of Nrf2 is tightly regulated by its repressor, Kelch-like ECH-associated protein 1 (Keap1), an adaptor subunit of Cullin 3-based E3 ubiquitin ligase. Normally, the Keap1-Nrf2 complex in the cytoplasm is in an inhibitory state. The Keap1-Nrf2 complex breaks down when a cell experiences oxidative stress, and then the released Nrf2 is transported to the nucleus and binds to antioxidant-response elements (AREs) of gene promoters to transcribe a series of antioxidants and detoxifying enzymes [27]. Hence, the Keap1/Nrf2/ARE pathway functions as a crucial mediator for cells to resist oxidative stress and engage in therapeutic targeting, as reported in a recent study [28].

Nrf2 also plays a role in melanogenesis in normal human melanocytes, in which it negatively regulates melanogenesis by decreasing the expression of tyrosinase and TRP1 through modulating the phosphatidylinositol 3-kinase (PI3K)/Akt signaling pathway [29]. In the clinic, it was reported that higher Nrf2-dependent transcriptional activity is required to sustain the redox balance in vitiligo patients [30]. Based on genetic studies, *Nrf2* gene polymorphisms were associated with susceptibility to vitiligo in Han Chinese populations [31]. *Nrf2* and its downstream detoxification genes were found to be upregulated in epidermal skin lesions of subjects with vitiligo [32], and catalase protein expression and activity are low in the epidermis of patients with vitiligo compared to healthy controls [33].

The Nrf2/ARE pathway can also inhibit the progression of inflammation by regulating anti-inflammatory gene expression [34]. In addition, dynamic changes in intracellular ROS levels can regulate stem cell self-renewal and proliferation through Nrf2-dependent Notch signaling from human airway basal stem cells [35]. Further research should be confirmed in clinical studies.

In summary, as one of the effectors for ROS, Nrf2 modulates stem cell homeostasis, negatively regulates melanogenesis, and inhibits the progression of inflammation. Many antioxidant drugs targeting the Nrf2/ARE pathway were identified and clinically applied [36]. Some of these drugs, such as ginsenoside Rk1, are reported to protect melanocytes from oxidative stress induced by H_2_O_2_ [37], providing a potential treatment for vitiligo patients by targeted antioxidant therapy. We review antioxidant therapy below.

### 3.2. Cell Adhesion Defects

Increased oxidative stress and adhesion defects in the epidermis are closely related. Accumulated ROS can cause protein oxidation and also lipid peroxidation, thus impairing cellular functions. Detachment of melanocytes due to oxidative stress was detected at the borders of vitiliginous lesions [38]. Interestingly, oxidative stress is more abundant in non-segmental vitiligo (NSV) than in segmental vitiligo (SV), as detected by lipid peroxidation using anti-4-HNE antibody [15,39]. In addition, epidermal structures are different, with NSV epidermis presenting acanthosis and SV showing an increase in rete ridges compared to healthy controls [40]. These observations suggest that the microenvironments of melanocytes in the skin of NSV and SV patients differ, even before depigmentation. These differences could impact the development and severity of the disease, which tends to be more extensive and unpredictable in NSV patients, but limited and rapid in SV patients [39]. Among adhesion-related proteins, E-cadherin is viewed as the major mediator of melanocyte-keratinocyte adhesion in human skin. The membranous distribution of E-cadherin is altered in both melanocytes and keratinocytes of vitiligo patients. Low E-cadherin levels are observed in melanocytes but not in keratinocytes of normal pigmented skin and in keratinocytes of depigmented skin of vitiligo patients [38,41,42]. These observations are coherent with the decline of E-cadherin in melanocytes as an early event in vitiligo pathogenesis. A decrease in E-cadherin may also induce activation of the Nrf2 antioxidant [32], the activation of which is also related in vitiligo patients. Matrix metalloproteinase (MMP)-9, an enzyme engaged in remodeling of the extracellular matrix (ECM), plays a vital role in response to interferon (IFN)-γ and tumor necrosis factor (TNF)-α, cytokines that are characteristic of vitiligo [43]. IFN-γ and TNF-α induce melanocyte detachment through disruption of E-cadherin and release of the soluble form of MMP-9 [44]. Dickkopf-related protein 1 (DKK1), an inhibitor of WNT signaling that regulates cell adhesion and migration, is overexpressed in the dermis of patients with vitiligo [45]. Fibronectin, a key protein in intercellular communication by binding to integrins, is highly expressed in lesional skin of vitiligo patients [46]. ICAM-1, a glycoprotein which enhances T-cell melanocyte attachment probably induced by TNF-α and IFN-γ [47], showed higher expression in vitiligo patients [48,49], which could represent links between cytokines and T-cell involvement in the pathogenesis of vitiligo. In summary, adhesion defects and detachment of melanocytes caused by extra ROS precede clinical manifestations of vitiligo.

### 3.3. Cell Death

In response to extra ROS accumulation, melanocytes undergo cell death in various ways including apoptosis, necrosis, autophagy, and ferroptosis [50].

(1) Apoptosis, the best characterized form of programmed cell death, is mediated by the ratio between antiapoptotic (Bcl-2) and proapoptotic (Bax) proteins, and an interaction of Fas and the Fas ligand. It was reported that there are a low Bcl-2/Bax ratio and elevated Fas levels in vitiliginous melanocytes [51,52], suggesting that vitiligo might be related to dysregulation of molecules regulating apoptosis. Cytotoxic T lymphocytes can induce cytotoxicity in melanocytes directly by releasing mediators such as TNF-α and IFN-γ. When IFN-γ binds to its receptor, the Janus kinase (JAK)/signal transducers and activators of transcription (STAT) pathway is activated and stimulates the transcription of the chemokine ligands CXCL9 and CXCL10 [53]. This creates a positive feedback loop for T-cell recruitment and function [53]. The levels of IFN-γ were reported to be elevated in lesional skin of patients with vitiligo [54], which induced an increase of CXCL10 in vitiligo melanocytes [55]. Treatment with H_2_O_2_ was found to upregulate several members of TNF-α receptor genes in melanocytes [56]. Taken together, these findings suggest the involvement of the TNF-α/TNFR pathway and IFN-γ pathway in melanocyte apoptosis of vitiligo. Current studies have found that tyrosinase-related protein (TYRP-1) and cleavage nuclear membrane antigen Lamin A/C (Asp230) can translocate into the apoptotic bodies of melanocytes [57]. The formation of melanocyte apoptotic bodies in vitiligo is mediated by the cytoskeletal protein activation pathway and the JNK-related apoptosis pathway [57]. This study links melanocyte apoptosis and cellular immunity in vitiligo pathogenesis.

(2) Different from apoptosis, necrosis is a type of cell death that causes cytoplasmic swelling and cell membrane rupture. In non-lesional vitiliginous skin, small numbers of necrotic cells have been observed [58]. The percentage of necrotic cells from melanocytes increases when the concentration of H_2_O_2_ increases [59], supporting the notion that necrosis possibly induces autoimmunity in vitiligo.

(3) Autophagy is the degradation of the cytoplasmic organelles or cytosolic components within lysosomes. Autophagy can be activated in response to not only extracellular signals such as starvation and pathogen infection, but also integration with cellular stress responses including oxidative stress, ER stress, and UPR accumulation [60]. The altered responses of vitiligo melanocytes to stressors have been linked to a reduced autophagic flux due to dysfunctions of the Nrf2/p62 pathway and its downstream antioxidant and detoxifying enzymes [61], suggesting that autophagy may play a role in the pathogenesis of vitiligo. Interestingly, perilesional skin in patients showed increased autophagy in comparison to surrounding active lesions, as assessed by increased expression of autophagy-related proteins (autophagy-related gene5 (Atg5) and light chain 3 (LC3)), concurrently with the down-modulation of sequestosome 1 (SQSTM1/p62) [62]. These phenomena raise the hypothesis of a protective role for autophagy in vitiligo [63]. Further studies have showed that autophagy inhibition exacerbates adverse metabolic effects in non-lesional vitiligo cells [64], suggesting that autophagy may be an adaptive response to continuous metabolic alterations for a protective attempt to antagonize degenerative processes and to enhance survival in normal vitiligo skin. The exact role of this process in the disease remains unclear.

(4) Ferroptosis is an iron-dependent oxidative cell death that is morphologically characterized by increased levels of lipid peroxidation with reductions in mitochondrial cristae [65]. The core regulator in ferroptosis is glutathione peroxidase 4 (GPX4), a selenium-dependent enzyme that catalyzes lipid peroxide reduction [65]. Hundreds of genes involved in the process of ferroptosis are transcriptionally regulated by the Nrf2 antioxidant, including *GPX4* [65]. Several studies showed that GPX is decreased in both the serum and tissues of vitiligo patients [66], raising the possibility of ferroptosis being triggered in vitiliginous melanocytes.

Although there are various modes of cell death, we believe that different forms of cell death may share similarities and be closely connected, contributing to the outcome through their integrated effects. Recent investigations have shown that oxidative stress has a significant impact on the interplay of various cell death modes [13]. However, the interplay of these modes is complex, and further research is needed.

## 4. Oxidative Stress and Immunity

### 4.1. Innate Immunity

ROS can modulate a variety of immune and inflammatory molecules. After sensing oxidative stress signals, innate immunity is quickly activated in response to those signals. Innate immunity then translates stress signals into proinflammatory signals, which provide a rapid response for better protection. Toll-like receptors and nucleotide oligomerization domain (NOD)-like receptors are related to proinflammatory signals. Autoantigens are produced from the body’s own cells, rather than being sourced from pathogen-associated molecules. They produce proinflammatory signals to induce an immune response (Table 1). Possible causes of autoantigen presentation originate from the following events: an unfolded protein response (UPR), 70-kDa heat shock protein (Hsp70) secretion, the exosome senescence-associated secretory phenotype (SASP), and cell death (Figure 2).

#### 4.1.1. Unfolded Protein Response (UPR)

When oxidative stress disrupts the redox potential, unfolded proteins form and accumulate. The UPR is a cellular restoration of homeostasis in response to damaged proteins and inappropriate protein folding from perturbed ER function. The UPR is an initial prosurvival signal cascade, in which translation is attenuated and the cell cycle is arrested to prevent further translational loading of the ER [67].

The activation of the UPR involves three principal ER transmembrane receptors: eukaryotic initiation factor 2α (eIF2α) kinase (PERK), activating transcription factor 6 (ATF6), and inositol requiring 1 (IRE1α) [68,69]. PERK is a type I transmembrane protein which phosphorylates eukaryotic translation initiation factor 2α (eIF2α) at serine 51 to inhibit protein synthesis [70]. ATF6 is a protein that activates gene transcription when unfolded proteins accumulate, and it is cleaved after being transported to the Golgi complex. The cleaved section of ATF6 was translocated to the nucleus, where it activates gene transcription of target genes [71]. IRE1α is a type I transmembrane protein with a protein serine/threonine kinase and endoribonuclease domain. It is activated when the ER chaperone BiP is released due to an accumulation of unfolded proteins. Once IRE1α is activated, it initiates splicing of the mRNA encoding XBP-1 in mammals, which activates the production of pro-inflammatory cytokines in macrophages [72]. IRE1α/XBP1s also regulates the homeostasis and survival of CD8α^+^ conventional dendritic cells [73]. Hence, UPR plays a vital role in inflammation and immune regulation [69].

The UPR is considered one of the crucial initiators in the pathogenesis of vitiligo. In addition to its function in antioxidant defense, UPR activation can promote the release of UPR-regulated chemokines that trigger an immune response against melanocytes [30] (Figure 2). Activation of the UPR increases the expression of the cytokines interleukin (IL)-6 and IL-8 by stressed melanocytes [74] and the release of C-X-C chemokine ligand (CXCL) [27,28,34], which mediate T-cell homing to the skin in vitiligo [75], and the production of CXCL16 from stressed keratinocytes that recruits cluster of differentiation 8-positive (CD8^+^) T cells [76]. Thus, dysfunctional activation of the UPR in melanocytes may cause more immunogenesis, which induces an attack on melanocytes in vitiligo.

#### 4.1.2. Hsp70

Another cytoprotector involved in progressive depigmentation is Hsp70. In response to stresses, Hsp70 acts as a chaperone which binds melanocyte-specific melanosomal proteins that facilitate protein folding and transportation, as well as major histocompatibility complex (MHC) I/II loading [77]. The presence of Hsp70-chaperoned proteins boosts DC activation [78] and promotes antigen-specific CTL-related immune responses [79]. Hsp70 directly activates DCs, as indicated by upregulation of the maturation markers CD40, CD83, and CD86. Hsp70 is believed to bind multiple receptor molecules on DC membranes including Toll-like receptor 4 (TLR4), TLR2, CD14, CD91, and CD40. Activation of DCs induced by Hsp70 and its receptor CD91 elevates expression of the TNF-related apoptosis-inducing ligand, which finally leads to skin homing of T cell migration [78,80,81]. Through activation of DCs, Hsp70 downregulates macrophage activity that causes inhibition of regulatory T cells (T_regs_) and supports T helper 17 (Th17)-mediated autoimmunity [78,82].

Overexpression of Hsp70 was observed in progressive depigmentation in experimental animal models [83,84,85,86]. Interferon (IFN) signaling is an important factor in the pathogenesis of vitiligo [87]. It was reported that Hsp70 can positively enhance the IFN signaling loop in the following manners. First, enhanced IFN-γ signaling from perilesional CTLs leads to upregulation of Hsp70, causing a positive feedback loop of Hsp70/CTL/IFNγ/Hsp70 [88]. Second, activation of DCs by Hsp70i facilitates IFN-α generation [89]. IFN-α induces the expression of CXCL9 and CXCL10 by keratinocytes and subsequently recruits C-X-C chemokine receptor 3-positive (CXCR3^+^)CD8^+^ T cells, forming an inflammatory cascade. These loops amplify the process and exacerbate the pathogenesis of vitiligo. The effects of Hsp70 open a potential alternative treatment for vitiligo patients [84].

#### 4.1.3. Exosome Senescence-Associated Secretory Phenotype (SASP)

Stressed cells are often characterized by their ability to develop exosome components of a SASP, a proinflammatory complex that consists of a mixture of ECM proteases, growth factors, chemokines, and cytokines. Such exosome components create an inflammatory microenvironment that causes inflammation and recruits immune cells to clear stressed melanocyte cells. T cells require antigen-presenting MHC-I and MHC-II to activate the immune system. Exosomes released from stressed cells can directly present intracellular self-antigens to T cells or promote T-cell activation in the presence of naïve DCs [90]. In summary, immunocompetent exosomes exert a critical function in innate and adaptive immune processes.

#### 4.1.4. Cell Death Debris

Debris from cells that have died can be highly immunogenic and can act as autoantigens to induce downstream T-cell responses [91]. The uncontrolled release of autoantigens via swelling of cytoplasmic organelles in necrotic cells leads to activation of immune responses and inflammation. Activation of apoptosis triggered by the Fas/Fas ligand interaction can be mediated by molecules which are involved in the pathology of vitiligo including CD8^+^ T cells, the TNF-α/TNR receptor (TNFR) pathway, the IFN-γ signaling pathway, and the CXCL10/CXCR3B pathway [52,55,92]. Therefore, the effective removal of cell death debris is necessary to prevent autoimmune reactions.

**Table 1 cells-12-00936-t001:** Oxidative stress-induced autoantigen and autoimmunity.

Stress Signal	Autoantigen	Innate Immunity	Adaptive Immunity	References
UPR	accumulation of unfolded or misfolded proteins that stimulate IL-6,IL-8, IL-11, tumor necrosis factor	macrophage	stimulates T lymphocytes	[73]
UPR (PERK-eIF2a and IRE1a-XBP1)	accumulation of unfolded or misfolded proteins that stimulate CXCL9,10,16	---	activate T cells, induced CXCR6^+^CD8^+^ T cells trafficking	[75]
activation of chaperone	Hsp70	dendritic cells, monocytes, macrophages	inhibit T_reg_ cells, activate Th17 cells	[81]
	Hsp70	langerhans cells	migration of skin hommg T cells	[85]
Exosome Senescence-Associated Secretory Phenotype (SASP)	ECM proteases, growth factors, chemokines, and cytokines	macrophage, dendritic cells	antigen-presenting MHC-I and MHC-II to activate T cells	[89]
cell death	cell death debris	dendritic cells	activate CD8^+^ and CD4^+^T cells	[90]

### 4.2. Adaptive Immunity

The concept that innate immunity can mediate the presentation of autoantigens to trigger autoimmunity bridges the gap between oxidative stress and adaptive immunity [3]. Melanocyte-specific CD8^+^ T cells were found to mediate the destruction of melanocytes in vitiligo [93]. Antigenic proteins from stressed melanocytes are carried by DCs and are specifically recognized by infiltrating T cells. In the leading edge of vitiligo depigmentation, patchy infiltration of T cells occurs. The serum frequency of cytosolic CD8^+^ T cells is higher in patients with vitiligo compared to healthy controls, and the frequency is related to disease severity. CD8^+^ T cells produce several cytokines such as TNF-α and IFN-γ that are primarily involved in melanocyte destruction. After binding of the IFN-γ proinflammatory cytokine to its receptor, the Janus kinase (JAK)/signal transducers and activators of transcription (STAT) pathway is activated and stimulates the transcription of the chemokine ligands CXCL9 and CXCL10 [53]. In the clinic, serum CXCL10 is representative of a specific biomarker that can monitor vitiligo activity and severity [94]. CXCL9/10 drives the recruitment of melanocyte-specific CD8^+^ T cell migration by interacting with CXCR3. In summary, the effect of the IFN-γ–CXCL9/CXCL10–CXCR3 axis on the killing of melanocytes by CD8^+^ T cells is significant.

T_reg_ cells are a subpopulation of T cells with tolerance to self-antigens, and the balance of T_reg_ cells is associated with disruption of autoimmune tolerance in vitiligo [95]. Based on current studies from single-cell RNA sequencing of human vitiligo, researchers revealed that CCL5-CCR5 cytokine signaling serves as a chemokine circuit between T_reg_ cells and effector CD8^+^ T cells [96]. The transcription factor Forkhead box p3 (FoxP3) is known to be expressed by T_reg_ cells and is associated with suppression of T-cell functions through downregulation of T-cell activation and cytokine genes while upregulating immunosuppressive cell-surface molecules such as CTLA-4 and CD25^+^ [97]. The concentration of FoxP3 is significantly reduced in lesional skin in vitiligo [98]. In conclusion, excess ROS can increase attacks on melanocytes through both innate and adaptive immunity.

## 5. Therapeutic Approaches to Ameliorate Oxidative Stress in Vitiligo Patients

Vitiligo is a chronic depigmenting skin disorder without effective treatments. Current vitiligo treatments use topical steroids or calcineurin inhibitors, often combined with phototherapy for better results. However, around 40% of patients experience a relapse within a year after stopping treatment [99]. Recently, new cell-based and cell-free regenerative approaches have been proposed as alternative treatments for vitiligo [53,100,101] These innovative regenerative approaches may help regenerate melanocytes and balance T-cell subsets, and are currently being tested.

In a randomized trial, objective vitiligo repigmentation obtained with topical catalase/dismutase superoxide (C/DSO) treatment is similar to topical 0.05% betamethasone [102], highlighting the need for better antioxidant medications. Our primary task is to find effective therapies to ameliorate oxidative stress to prevent destruction of melanocytes in vitiligo patients. Aside from phototherapy and steroid treatment, alternative treatments that target antioxidant activity in vitro and/or in vivo are emerging [103]. We summarize potential therapeutic approaches to reduce oxidative stress in vitiligo in Table 2. Polyphenols are beneficial plant compounds with antioxidant properties. *Gingko biloba* extract (EGB 761) was studied as a free radical scavenger to boost survival of skin flaps in rats [104]. Clinical trials (ClinicalTrials.gov Identifier: NCT00907062) and research have shown that patients with stable vitiligo can achieve repigmentation after taking EGB761 [105]. Other polyphenol compounds including apigenin, baicalin, and quercetin, were documented to activate the Nrf2 pathway [106,107,108,109,110]. Several organic substances (simvastatin, aspirin, vitamin D, and minocycline), as well as herbal medicines (glycyrrhizin, afzelin, 6-shogaol, geniposide, 8-Methoxypsoralen, and cinnamaldehyde), were documented to protect human melanocytes against oxidative stress through activation of the Nrf2/ARE pathway [111,112,113,114,115,116,117,118,119,120,121,122,123]. Of note, adipose tissue secretomes have the ability to counteract oxidative stress by physiologically stimulating intracellular antioxidant enzymes [124]. In addition to immunomodulation [53], mesenchymal stem cells (MSCs) from the dermis can regulate melanocyte proliferation and apoptosis by targeting the PTEN/PI3K/Akt pathway [101,125]. Inorganic vitiligo therapies such as molecular hydrogen, palladium, and platinum nanoparticles, provide protection of melanocytes from oxidative stress by activating Nrf2/ARE signaling [126,127].

The antioxidants mentioned above are still far from being therapeutically adopted because of small numbers of participants, poor operability, and heterogeneity in patients. With emerging innovative strategies for drug delivery that allow antioxidants to permeate into deeper skin layers [128], it is worth implementing standardized compatibilities of antioxidants in large clinical trials in the future.

## 6. Conclusions and Perspectives

ROS play crucial roles in the initiation of vitiligo. Accumulating ROS cause melanocyte destruction and dysfunction from several aspects, including ER stress, mitochondrial dysfunction, lipid peroxidation, and adhesion defects. Antioxidants such as the Nrf2/ARE pathway and the Hsp70 chaperone form the first defense in preventing stressed melanocytes from undergoing cell death. Otherwise, the death of cells, either from necroptosis or ferroptosis, can be highly immunogenic for generating an inflammatory microenvironment. Innate immunity triggered by the presence of autoantigens from stressed melanocytes links the immune mechanism from oxidative stress to adaptive immunity. In the case of vitiligo, for example, the activation and maturation of DCs through Hsp70 presents autoantigens, and these activate CD8^+^ T cells to destroy melanocytes. Growing attention to T_reg_ cells failed to suppress autoreactive immunity, while CD8^+^ CTLs kill melanocytes with the support of Th17 cells.

Although anti-ROS treatments have shown potential as therapy for vitiligo, their use is currently restricted by concerns surrounding specificity, efficacy, safety, and lack of long-term data. Inferring from events during the pathology of vitiligo, emerging therapeutic options originate from activation of the Nrf2/ARE pathway, Hsp70 mutations, inhibition of adaptive immunity-related autoantigens, replenishing of T_reg_ abundances, and blocking of T helper cells. In addition, ferroptosis is currently reported to be involved in epidermal melanocyte destruction in vitiligo [103], even though its mechanism remains unclear. Further studies focused on these issues may provide references for new drug development and treatment options for vitiligo.

## Figures and Tables

**Figure 1 cells-12-00936-f001:**
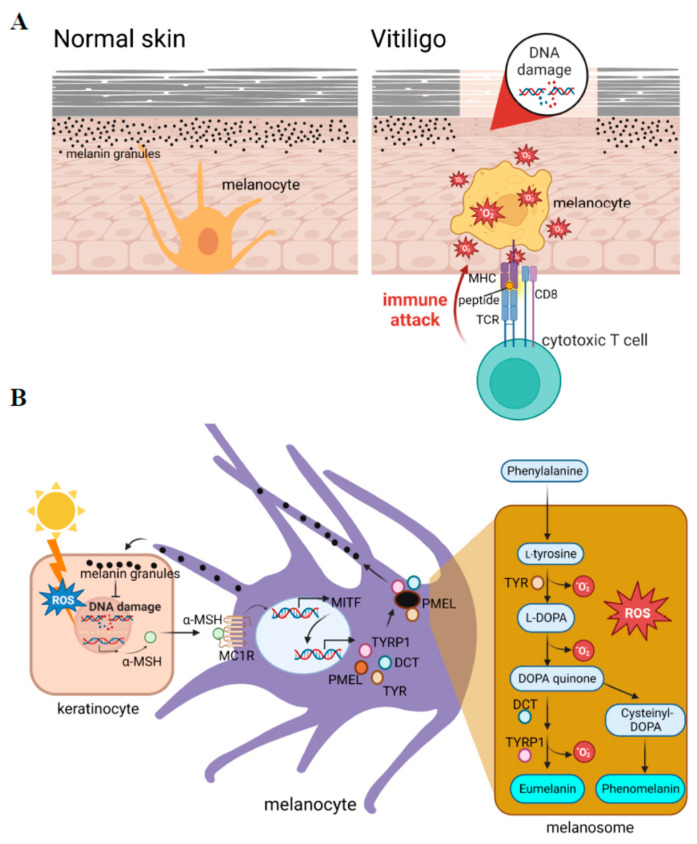
Reactive oxygen species (ROS) play crucial roles in the initiation of vitiligo. (**A**) Vitiligo is a depigmentation skin disorder caused by dysfunctional melanocytes that are attacked by immune cells. (**B**) Melanogenesis generates intracellular ROS. The production of melanin is activated by exposure to sunlight, which leads to the production of the ligand for activation of the transcription of microphthalmia-associated transcription factor (MITF) in melanocytes. MITF activates a series of pigment-specific enzyme genes including *TYR* (tyrosinase), *DCT* (dopachrome tautomerase), and *TYRP1* (tyrosinase-related protein 1). The process of melanogenesis is shown in the right penal, in which a pro-oxidant state is generated during melanogenesis. α-MSH: alpha-melanocyte-stimulating hormone; PMEL: pre-melanosome protein; L-DOPA: 3,4-dihydroxy L-phenylalanine.

**Figure 2 cells-12-00936-f002:**
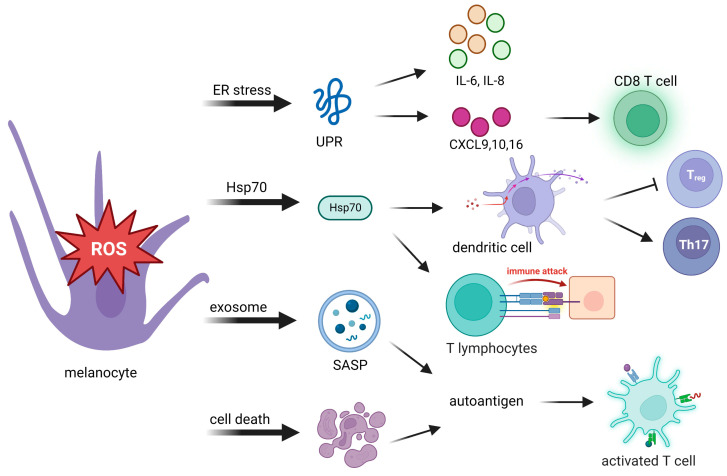
Oxidative stress activates adaptive immunity through autoantigen presentation to innate immunity. Stressed melanocytes produce autoantigens in the manner of cytokines, extracellular exosomes, and cell death debris. Autoantigens can be highly immunogenic, which activates cytotoxic T cells, regulatory T (T_reg_) cells, and T helper cells. UPR, unfolded protein response; IL-6, interleukin-6; IL-8, interleukin-8; CXCL, C-X-C chemokine ligand; Hsp70, 70-kDa heat shock protein; SASP, senescence-associated secretory phenotype; Th17: T helper 17.

**Table 2 cells-12-00936-t002:** Potential therapeutic approaches targeting antioxidant activity.

Class	Antioxidants	Target	Mechanisms	NCT No. of Clinical Trials	Status	References
polyphenols	*ginkgo biloba* extract (EGB761)	Nrf2 pathway	activate the Nrf2 pathway, promotes the expression of the antioxidant genes heme oxygenase (HO)-1 and SOD	NCT01006421 NCT00907062 (Phase I)	repigmentation more than 50% from baseline as the primary outcome (time frame: 3, 6 and 9 months)	[104,105]
	apigenin	Nrf2 pathway	activate the Nrf2 pathway	---	---	[106]
	baicalin	Nrf2 pathway	activate the Nrf2 pathway	---	---	[107,108]
	quercetin	PI3K/Akt and p38 pathway	anti-inflammatory and antioxidative capacity, protect human melanocyte from H_2_O_2_-induced apoptosis	---	---	[109]
	chalcones	PI3K/Akt and GSK3β pathway	activate PI3K/Akt and GSK3β pathways, promote the formation of epidermal melanin, induce the recoloration of vitiligo	---	---	[110]
organics	glycyrrhizin	Nrf2 pathway	activates Nrf2 and induces the expression of heme oxygenase (HO)-1 in macrophages	---	---	[111]
	afzelin	Nrf2 pathway	activates Nrf2 and induces the expression of heme oxygenase (HO)-1 and catalase	---	---	[112]
	6-shogaol	Nrf2 pathway	upregulating the mRNA expression of the antioxidant enzyme HO-1, and protein expression of Nrf2, NAD(P)H: quinine oxidoreductase 1 (Nqo1)	---	---	[113]
	simvastatin	Nrf2 pathway	activates the Nrf2 pathway	NCT01517893 (Phase II) NCT03247400 (Phase II)	[NCT01517893] participants with 33% decrease in the Vitiligo Area Scoring Index (VASI) [NCT03247400] reduction in lesional skin area	[114]
	aspirin	Nrf2 pathway	induce Nrf2 nuclear translocation, induce HO-1 expression in human melanocytes, reduction in the acute serum immunologic markers of T cell activation in non-segmental vitiligo	---	---	[119]
	geniposide	PI3K/Akt pathway	PI3K inhibitor LY294002 inhibit cell viability, apoptosis, and antioxidant enzyme activity	---	---	[116,117]
	8-Methoxypsoralen	Akt pathway	reduce AKTphosphorylation, scavenge oxygen free radicals	---	---	[118]
	vitamin D	Wnt/β-catenin pathway	activate the Wnt-β catenin	NCT05364567 NCT04872257	---	[120,121]
	cinnamaldehyde	AhR pathway	inhibit abnormal activation of the AhR pathway, reduce the production of ROS	---	---	[122]
	minocycline	JNK, and p38 MAPK pathway	inhibit the activation of JNK, p38 MAPK, and caspase 3 induced by H_2_O_2_, prevent the loss of melanoctres	---	---	[123]
cell-based and cell-free	adipose tissue secretome	Wnt/β-catenin pathway	promotes glycogen synthase kinase 3β inactivation and consequently Wnt/β-catenin pathway activation	---	---	[124]
	mesenchymal stem cells (MSCs)	PTEN/PI3K/Akt pathway	regulate melanocyte proliferation, immunomodulation	NCT03013049 (Dermal-MSCs)	>90% repigmentation from stable vitiligo patients	[101,125]
inorganics	hydrogen	Nrf2 pathway	Wnt/β-catenin–mediated activation of Nrf2 signaling, reduced intracellular ROS accumulation,	---	---	[126]
	palladium and platinum nanoparticle	Nrf2 pathway	activates aryl hydrocarbon receptor and Nrf2	---	---	[127]

## Data Availability

Not applicable.

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
