# Peer review of "The Role of Oxidative Stress in Vitiligo: An Update on Its Pathogenesis and Therapeutic Implications"

_cells, 2023, doi:10.3390/cells12060936_

Round 1

Reviewer 1 Report

The review "The Role of Oxidative Stress in Vitiligo: An Update on Its Pathogenesis and Therapeutic Implications" aims to present the importance of oxidative stress in the pathogenesis of vitiligo. Vitiligo is the most common depigmenting disorder and, despite therapeutic advances, no treatment achieves complete repigmentation. It is therefore essential to understand the cause of the disease to develop effective therapeutic strategies. The etiology of vitiligo is multifactorial and involves genetic factors, oxidative stress and autoimmunity. The authors focus on oxidative stress which is considered to play a crucial role in the etiology of vitiligo.

In general, the study is clearly presented and gives an overview of the origin and consequences of oxidative stress in the context of vitiligo. However some aspects should be improved, here is the list of comments.

-Lines 87-101: The authors mention the generation of ROS during melanin synthesis but do not mention the role of the melanosome as a protective organelle in the cell. This point needs to be discussed.

-Figure 1: In the legend the description of the elements of the figures is missing, e.g. black dots, membrane proteins on the surface of T cells and melanocytes… Typos should be corrected (right penal…). Why is the pigment only located in the stratum corneum?

-Line 84: “A series of cellular stressors…” be more precise, how does proliferation and differentiation produce ROS?

-Line 138: It is important to mention whether the data in the cited literature are from studies of vitiligo epidermis or from other biological systems (e.g. ref 17). This will avoid erroneous (or not yet proven) conclusions about the pathology of vitiligo.

-Lines 147-163: It is worth mentioning that in both melanocytes and keratinocytes, the membranous distribution of E-cadherin is altered. Specifically, in melanocytes (not in keratinocytes) of normal pigmented skin and in keratinocytes of depigmented skin of vitiligo patients (Wagner et al., 2015 doi:10.1038/jid.2015.25; Kim and Lee, 2010 doi: 10.1038/jid.2010.99). This may help to better understand the sequential alterations found in the vitiligo epidermis. It would also be interesting to mention if oxidative stress is detected in the different types of vitiligo (see Delmas and Larue 2019 doi: 10.1111/exd.13858).

-Lines 166-183: A conclusion is missing in the paragraph of cell death. Also, could the authors be more specific about the vitiligo samples in which the various cell deaths were observed? (in stable or active, generalized or segmental vitiligo, in normal pigmented skin, perilesional or lesional …). What is the frequency of the different cell death? The authors should discuss the role of autoantigens released from apoptotic melanocytes under oxidative stress (see Tian et al. 2021 doi:10.1155/2021/7617839).

- Lines 290-316: It will be useful to know for each component mentioned, whether  the therapeutic approaches are effective in reducing oxidative stress in patients with vitiligo and for how long and with what efficacy repigmentation has been achieved. This could be accomplished by adding additional columns to Table 1.

- The authors should discuss more generally what the role of treatments targeting oxidative stress would be in the context of currently proposed therapies (NB-UVB phototherapy, topical corticosteroids and calcineurin inhibitors ..) or new therapies such as cell-based and cell-free regenerative approaches (see Bellei et al. 2022 doi:10.3390/biomedicines10112744).

. The authors should discuss and cite at least one randomized trial “Sanclemente et al.. 2008 doi: 10.1111/j.1468-3083.2008.02839.x” which compared the effect of topical 0.05% betamethasone versus topical catalase/dismutase superoxide.

- The authors primarily discuss antioxidant therapy targeting the Nrf2 pathway and should also discuss antioxidants targeting the PI3K/Akt pathway, the aromatic hydrocarbon receptor (AhR) pathway and others.

-The authors should also propose methods to detect oxidative stress in patients for patient management.

Reviewer 2 Report

In the present review, authors have discussed on the role of oxidative stress in vitiligo pathogenesis. However, the review article is very concise; the therapeutic implications of oxidative stress related aspect did not elaborate by authors. There are a few points that should be taken into consideration for improvement of the manuscript.

11. As presented in the title, the review article lacks the therapeutic implication of oxidative stress. Only a small paragraph is presented and author should also review the antioxidant therapies and other molecular therapies based on oxidative/antioxidative status.

22. Under the cell adhesion defects, authors must also discuss about ICAM molecules.

33. Cell death: Authors forget to discuss about the role of TNF-α and IFN-γ in the apoptosis and their induction by ROS.

44. Authors have mainly focused on Nrf2/ARE pathway and hence the title of the manuscript should be changed.

55. Authors should also discuss about the personalized medicine based on oxidative status of the patients.

66. Authors must discuss about Pseudocatalase and herbal medicines used for combating the oxidative stress in vitiligo patients.

77. Authors should give brief, concise and constructive introduction of vitiligo mentioning the prevalence of vitiligo.

88. Authors should also include the oxidative stress mediated autophagy of melanocytes.

99. Authors must also discuss the role of mitochondrial role in oxidative stress, in context of vitiligo.

110.  Moreover, mitochondrial oxidation happens in all cell types. How author will justify it with vulnerability of melanocytes with oxidative stress?

111.  GSH plays a pivotal role in maintaining a balanced antioxidant: oxidant; authors should also include and explain in the review article.

112.  Authors did not discuss the oxidative stress related pathogensis of vitiligo based on disease severity/activity (Active vitiligo, stable vitiligo, mild or severe vitiligo)

113.  Authors should also mention the limitations of the study at the end of the manuscript.

114.  Authors must also discuss the previous studies carried out in vitiliginous skin and melanocytes; specifically for the high oxidative stress and altered antioxidant status.

115.  Authors must present the status of various oxidants/antioxidants studied in vitiligo in tabular form (for the analyzed molecules in blood/skin) for the ease of the readers.

116.  Authors must also describe how oxidative stress can lead to autoimmunity (the various mechanisms).

117.  Authors have avoided many important previous findings related of the antioxidant status and oxidative stress related studies.

118.  Authors should critically go through the entire manuscript and correct typographical and grammatical mistakes.

Reviewer 3 Report

In the present review, Chang and Ko have reviewed literature to emphasise on the role of oxidative stress in vitiligo. Authors here also discusses various factors responsible for oxidative stress, their effect and implications in vitiligo. Over all, the article is nicely written however several points have to be taken into the consideration for publication of this article in a final form.

1.    Authors should provide references for statements in paragraph 2 of the Introduction section.

2.    Page 2, lines 42-44. The statement overproduction… tissue damage is too vague. Authors should specifically discuss about pathological situations, genetic predisposition, cell and tissue damage which they refer to, along with citing supporting literature.

3.    Figure 1 legend. Statement ‘Immunity is …signaling’ (lines 50-52) is not required as the figure 1A doesn’t depict ROS mediated immune activation.

4.    Line 66. ‘UPS’ should be corrected as ‘UPR’.

5.    The statements in this review article are poorly supported by relevant references. All the informative sentences must be supported with relevant references except for those which are authors’ own suggestions, hypotheses or thoughts.

6.    Line 87, the statement ‘melanosomes are intracellular structures that produce ROS’ is misleading. ROS production is not a primary function of melanosomes. Authors should correct the statement.

7.    I do not agree with the definition and possible causes of ‘autoantigen presentation’ mentioned in line 190-195. Authors should be elaborate the why they think autoantigens are generated following stress signals and how the mentioned factors involve

8.     The role of oxidative stress and UPR (Section4.1.1) could be elaborated and UPR pathways involved should be discussed in more details. Authors should integrate recent literature in this section. For ex. PMID: 33613564.

9.    It may be useful to provide a table of autoantigens discovered in vitiligo till date in section 4.2.

Round 2

Reviewer 2 Report

The manuscript has been improved now. However, still few minor corrections are needed, as suggested below.

1.       ICAM-1 is also induced by IFN-gamma and has been shown to be increased in vitiligo patients. Authors must discuss this. [https://www.liebertpub.com/doi/abs/10.1089/jir.2012.0171]

2.       For discussing the mitochondrial role in oxidative stress, in context of vitiligo, author must cite and discuss the below mentioned article involving the SOD2 (mitochondrial SOD). [https://doi.org/10.1016/j.freeradbiomed.2013.08.189]

3.       “Authors must also describe how oxidative stress can lead to autoimmunity.” For this authors can cite the popular article. [https://onlinelibrary.wiley.com/doi/full/10.1111/exd.12103]

Reviewer 3 Report

The authors have addressed most of my earlier suggestions and improved the manuscript except for comments No. 7 and 9. I think there is some confusion in the authors' understanding of autoantigens, which can be also stipulated in the second column of Table 1. CXCL16 is a cytokine belonging to the chemokine family, not an autoantigen. Similarly, IL6, IL8, etc.. are not autoantigens. They are the factors or players of immune response but not autoantigens.

Autoantigens are self-proteins (Not produced unexpectedly in the body) that aberrantly activate the body's own immune response either by cellular, humoral, or both mechanisms.

The authors should correct this serious concern in the revised manuscript.
